# New Methodology for the Identification of Metabolites of Saccharides and Cyclitols by Off-Line EC-MALDI-TOF-MS

**DOI:** 10.3390/ijms21155265

**Published:** 2020-07-24

**Authors:** Gulyaim Sagandykova, Justyna Walczak-Skierska, Fernanda Monedeiro, Paweł Pomastowski, Bogusław Buszewski

**Affiliations:** 1Centre for Modern Interdisciplinary Technologies Nicolaus Copernicus University in Toruń, Wileńska 4, 87-100 Toruń, Poland; sagandykova.gulyaim1@gmail.com (G.S.); walczak-justyna@wp.pl (J.W.-S.); fernandamonedeiro@usp.br (F.M.); pawel_pomastowski@wp.pl (P.P.); 2Department of Environmental Chemistry and Bioanalytics, Faculty of Chemistry, Nicolaus Copernicus University in Torun, Gagarina 7, 87-100 Toruń, Poland

**Keywords:** biologically active compounds, saccharides, cyclitols, electrochemistry, MALDI-TOF-MS

## Abstract

A combination of electrochemistry (EC) and matrix-assisted laser desorption/ionization time-of-flight mass spectrometry (*off-line* EC-MALDI-TOF-MS) was applied for determination of the studied biologically active compounds (D-glucose, D-fructose, D-galactose, D-pinitol, L-*chiro*-inositol, and *myo*-inositol) and their possible electrochemical metabolites. In this work, boron-doped diamond electrode (BDD) was used as a working electrode. MALDI-TOF-MS experiments were carried out (both in positive and negative ion modes and using two matrices) to identify the structures of electrochemical products. This was one of the first applications of the EC system for the generation of electrochemical products produced from saccharides and cyclitols. Moreover, exploratory data analysis approaches (correlation networks, hierarchical cluster analysis, weighted plots) were used in order to present differences/similarities between the obtained spectra, regarding the class of analyzed compounds, ionization modes, and used matrices. This work presents the investigation and comparison of fragmentation patterns of sugars, cyclitols, and their respective products generated through the electrochemistry (EC) process.

## 1. Introduction

Carbohydrates are a complex group of organic compounds occurring in all plants. They are the main source of energy for both humans and plant organisms. They play an important role in the proper functioning of the brain [1,2]. Cyclitols are another group of compounds found in plant material. Cyclitols are responsible for the plant’s self-protection against unfavorable environmental conditions, they are involved in signal transduction, in biogenesis and osmoregulation, and in phosphate storage [1,3,4,5,6].

Biologically active substances and compounds with potential biological activity of plant origin, i.e., both those with beneficial effects and used as medicines and those that have harmful effects on human health, require monitoring in the environment and in the human body [7]. One of the techniques that allows the monitoring of biologically active compounds is matrix-assisted laser desorption/ionization time-of-flight mass spectrometry (MALDI-TOF-MS) [8]. The MALDI-TOF-MS technique in recent decades became irreplaceable in the analysis of individual compounds and their mixtures and also in complex biological matrices. The development of ‘soft’ ionization methods, high-throughput techniques and quantitative methods has expanded the use of mass spectrometry (MS) in the study of the structure, function, and modification of low molecular weight compounds [8,9]. The use of MALDI-TOF-MS in the analysis of metabolites of low molecular weight compounds allows eliminating chromatographic separation [10]. MALDI-MS is widely used for the analysis of large compounds, such as proteins, peptides, and also whole bacteria cells [11,12,13]. Analysis of low molecular weight compounds by MALDI-MS is a current analytical challenge [14]. However, despite many limitations such as signal suppression, acidity of the matrix solution, and degradation of the analyte, MALDI-MS has found its place in application in the analysis of low molecular weight compounds [14,15]. MALDI-TOF-MS was used in the analysis of anthocyanins, sugars, and organic acids in strawberry [16], the analysis and characterization of oligosaccharides [17], and the identification of flavonoids in berry extracts [18] as well as sugars and alkaloids in food [19].

In recent years, electrochemistry (EC) has been increasingly used in research on the metabolic biotransformation of biologically active compounds. Electrochemical methods allow for real-time measurements often with high resolution in both in vitro and in vivo studies. The electrodes can be used to analyze metabolites released by the cell or to assess the consumption of compounds already in the culture medium, but they can also generate compounds in the immediate vicinity of the cell and then measure how the cell responds to such stimulation [20,21]. The combination of electrochemistry with mass spectrometry allows mimicking the reactions of phase I and II metabolism as well as identifying electrochemical products that can serve as potential metabolites [22,23]. This simple instrumental analytical technique allows the production, detection, and identification of a number of metabolic products, including reactive intermediate metabolites, that are responsible for drug activation. EC/MS also allows the identification of the final products of electrochemical reactions of the tested compounds and is used in redox (oxidation–reduction) studies [24,25,26]. 

The study of a compound biotransformation is a complex process. Among the available analytical techniques, EC-MS is a universal tool that has been used to confirm the presence of parent compounds and identify their candidate metabolites, which are most often targeted in a complex biological environment in metabolomics studies. In this work, a new method using an *off-line* EC-MALDI-TOF-MS system for the analysis of possible products of low-molecular biologically active compounds was developed. For the first time, an electrochemical system and matrix-assisted laser desorption/ionization time-of-flight mass spectrometry were used to determine and identify electrochemical products of selected saccharides (D-glucose, D-fructose, and D-galactose) and cyclitols (D-pinitol, L-*chiro*-inositol, and *myo*-inositol). Statistical methods were used to provide data visualization and to highlight the main characteristics of data structure, supporting the discussion of fragmentation pathways. The present goals comprised the elucidation of fragmentation patterns of sugars, cyclitols, and their suggested metabolites, as well as the investigation of correlations between their MS spectra. The intention was to propose a wide range of EC products of these compounds, presenting a methodology that could be used in the monitoring of such species and the modeling of potential metabolic reactions of phase I oxidative cell metabolism.

## 2. Results and Discussion 

### 2.1. Effect of Matrix

MALDI-TOF-MS has been mostly used to identify large biomolecules such as proteins, peptides, nucleic acids, and polymers. Qualitative and quantitative analysis of low molecular weight compounds (LMWC) is challenging due to the large number of signals coming from the ionizing matrix, which consequently prevents identification of the analytes. Despite this limitation, the application of MALDI-MS in the analysis of low molecular weight compounds has been constantly increasing due to the large tolerance of the buffers used during the analysis, formation of mostly single-charged ions, high sensitivity, and throughput [15]. 

Two matrices, a benzoic acid derivative, 2,5-dihydroxybenzoic acid (DHB), and a cinnamic acid derivative, α-cyano-4-hydroxycinnamic acid (HCCA), were employed. Both matrices belong to the first-generation matrices with molecular weight below 300 Da [15]. The use of these matrices ensured stable ionization, high reproducibility, and a high resolution of recorded MS spectra. MALDI spectra for DHB and HCCA matrices were registered to eliminate the signals coming from matrices from the spectra. Several ions were formed during laser irradiation for the HCCA matrix, as it can be observed from the spectra. Except for a molecular ion [M + H]^+^ at *m*/*z* 190.0, a fragment of high intensity was probably formed at *m*/*z* 172.0 and *m*/*z* 379.0, which may correspond to [M-H_2_O + H]^+^ and matrix cluster [2M + H]^+^, respectively. Less intense fragments also could appear in the spectrum at *m*/*z* 146.1, *m*/*z* 164.1, *m*/*z* 212.0, and *m*/*z* 335.1, which may correspond to [M-CO_2_ + H]^+^, [MH-CN]^+^, sodium adduct [M + Na]^+^, and [2M + H-CO_2_]^+^, respectively [15,27]. In case of the DHB matrix, less ions coming from the matrix were generated. A signal of high intensity at *m*/*z* 137.0 that may correspond to [M+H-H_2_O]^+^ can be observed from the spectrum as well as less intense signals at *m*/*z* 155.0 and *m*/*z* 177.0 that may correspond to the protonated molecular ion from matrix [M + H]^+^ and the sodium adduct [M+Na]^+^, respectively. It can be assumed that the DHB matrix is found to be more suitable for the analysis of low molecular weight analytes such as sugars and cyclitols, since less ions coming from the matrix were generated during laser irradiation, which can cause interferences during the identification of analytes. 

### 2.2. Off-Line EC-MALDI-TOF-MS

In order to generate possible transformation products of cyclitols and saccharides, a system consisting of an off-line electrochemical cell coupled with matrix-assisted laser desorption/ionization was used. The off-line EC-MALDI system was used for the first time in the analysis of metabolites of low molecular weight compounds. The methodology of the experiment included use of the boron-doped diamond electrode (BDD) working electrode and ammonium acetate at an approximate pH of 7.4. Standards of cyclitols and saccharides were introduced to the electrochemical cells, and fractions after conversion were collected manually. Each fraction was evaporated using a Labconco Centri Vap DNA concentrator (Kansas City, USA). In a second step, fractions were analyzed by MALDI-TOF-MS. Figure 1, Figure 2, Figure 3, Figure 4, Figure 5 and Figure 6 present the suggested fragmentation pathways of cyclitols and saccharides before and after electrochemical analysis. There are very few reports regarding the analysis of cyclitols and saccharides using MALDI-TOF-MS in the literature. 

Moreover, representatives of species of sugars and cyclitols were selected in order to observe the possible similarities and differences in their EC products, since these analytes have similar structural features. Such similarity was interesting, since they possess opposite activities (e.g., in *Diabetes Mellitus*) that in particular cases is structure-dependent. In addition, the anti-diabetic activity of cyclitols has different aspects, but the mechanisms are still understood poorly. Besides that, literature on the potential metabolites of cyclitols in human organism is quite scarce [28].

#### 2.2.1. D-Pinitol

Figure 1a shows a suggested fragmentation pathway of D-pinitol (signals before the electrochemical process are denoted in red color, signals after the electrochemical process are denoted in blue). Figure 1b,c show the mass spectra of D-pinitol before and after electrochemical analysis. The signals A and B (in both mass spectra; denoted in black color) belong to the DHB matrix. In the spectrum, the molecular ion of D-pinitol at *m*/*z* 195.0 (C) can be observed. The signal at *m*/*z* = 193.0 (D) could be derived from the signal C by the loss of hydrogen (dehydrogenation). The ion at *m*/*z* = 212.1 (R) could be originated from electrochemical oxidation and the detachment of a methyl group from ion 193.0 (D). The signal *m*/*z* 228.0 (S) appeared only in the spectrum before the electrochemical process and may correspond to the addition of a hydroxyl group. Ions at *m*/*z* 230.9 (more intense before the electrochemical process and less intense after the electrochemical process; T) and *m*/*z* 249.9 (occurred only before electrochemical process; U) may correspond to the addition of two hydrogen atoms (hydrogenation) and the addition of sodium and loss of four hydrogen atoms, respectively. Other products of D-pinitol are E, J, and K (Figure 1a,c). The ions at *m*/*z* 370.9 (E), 354.9 (J), and 346.9 (K) may correspond to [C_14_H_26_O_11_]^+^ (dimer of D-pinitol), [C_13_H_22_O_11_]^+^ (loss of methyl group), and [C_13_H_14_O_11_]^+^ (loss of eight hydrogen atoms from hydroxyl groups). The fragment that was observed at *m*/*z* 392.9 (F) may correspond to [C_14_H_25_NaO_11_]^+^ (dimer of D-pinitol and adduct with sodium). The signals at *m*/*z* 376.9 [C_13_H_21_NaO_11_]^+^ (G) and *m*/*z* 361.9 [C_13_H_23_NaO_10_]^+^ (I) could result from the ion at *m*/*z* 392.9 by the loss of a methyl group (14 u) and oxygen atom (16 u), respectively. The ions detected at *m*/*z* 366.9 (H), 344.0 (L), 328.9 (M), 288.2 (N), 273.0 (O), and 265.9 (P) can be ascribed to [C_14_H_22_O_11_]^+^, [C_13_H_12_O_11_]^+^, [C_13_H_12_O_10_]^+^, [C_13_H_20_O_7_]^+^, [C_13_H_21_O_6_]^+^, and [C_13_H_13_O_6_]^+^, probably corresponding to the formation of a dimer of D-pinitol and the loss of two hydrogen atoms (2 u), loss of one carbon atom and 10 hydrogen atoms (22 u), loss of one oxygen atom (16 u), loss of 3 × O, addition of 8 × H, and the loss of oxygen (16 u), and the loss of five hydrogen atoms (5 u), respectively. The ions at *m*/*z* 273.0 (O) and *m*/*z* 265.9 (P) may correspond to the formation of radical cations (Figure 1a,b).

#### 2.2.2. L-*Chiro*-Inositol

The signal observed at *m*/*z* 164.1 (D) may correspond to the loss of hydroxyl group (17 u) from L-*chiro*-inositol (Figure 2b). The molecular ion [M + H]^+^ was not observed in the MS spectra (Figure 2b,c). The ion at *m*/*z* 193.0 (C) could be formed by the addition of a methyl group to the hydroxyl group. In Figure 2c, L-*chiro*-inositol’s derivative at *m*/*z* 199.0 [C_6_H_8_NaO_6_]^+^ (E) can be observed. This ion could be formed by replacement of the methyl group to sodium (Figure 2a). The fragment at *m*/*z* 230.9 [C_6_H_14_O_9_]^+^ (G) could be originated from the ion at *m*/*z* 214.9 [C_7_H_11_NaO_6_]^+^ (F) by the loss of CH–Na and the addition of three hydroxyl groups (51 u). The dimer of L-*chiro*-inositol can be observed at *m*/*z* 343.2 [2M]^+^ (H) presented in Figure 2a,b. Ions at *m*/*z* 304.2 [C_12_H_16_O_9_]^+^ (I), *m*/*z* 298.3 [C_12_H_10_O_9_]^+^ (J), *m*/*z* 256.2 [C_12_H_16_O_6_]^+^ (K), and *m*/*z* 244.2 [C_11_H_16_O_6_]^+^ (L) may correspond to the loss of two oxygen atoms and seven hydrogen atoms from the phenol group (H_7_O_2_, 39 u), the loss of six hydrogen atoms (dehydrogenation), the loss of three oxygen atoms (oxidation) and addition of six hydrogen atoms (hydrogenation), and the loss of one carbon atom (cleavage of ring), respectively. The ion at *m*/*z* 339.0 (M) can be ascribed to the adduct of one molecule of the DHB matrix with L-*chiro*-inositol and sodium (Figure 2c). The signal at *m*/*z* 317.0 (N) is created from the ion M by the loss of sodium and addition of one hydrogen atom. The ion at *m*/*z* 316.2 (O) may result from the dehydrogenation of a carboxyl group from DHB molecules at *m*/*z* 317.0. Another signal at *m*/*z* 288.2 (P) was observed in spectra before the electrochemical process and may correspond to [C_13_H_20_O_7_]^+^. The peak at *m*/*z* 284.3 (R) may correspond to the loss of four hydrogen atoms (dehydrogenation) from ion [C_13_H_20_O_7_]^+^. The ion at *m*/*z* 376.9 (S) could result from the formation of a sodium adduct of L-*chiro*-inositol with one molecule of the DHB matrix and two hydroxyl groups [C_13_H_21_NaO_11_]^+^. The ions detected at *m*/*z* 360.9 (T), 354.8 (U), 331.017 (V), and 332.2 (X) can be ascribed to [C_13_H_21_NaO_10_]^+^, [C_13_H_15_NaO_10_]^+^, [C_13_H_15_O_10_]^+^, and [C_13_H_16_O_10_]^+^, probably corresponding to the loss of one hydroxyl group (17 u) from the DHB ring, dehydrogenation (the loss of four hydrogen atoms), the loss of one sodium atom from the carboxyl group of DHB molecules, and hydrogenation (addition of one hydrogen), respectively. Furthermore, the ion at *m*/*z* 361.9 (Y) could result from the addition of one hydrogen atom to the ion at *m*/*z* 360.9. The ion at *m*/*z* 346.9 (W) may be derived from the ion at *m*/*z* 354.8 by the loss of eight hydrogen atoms.

#### 2.2.3. *Myo*-Inositol

In the case of *myo*-inositol, the molecular ion also could not be observed (Figure 3b,c). Signals at *m*/*z* 174.9 [C_6_H_6_O_6_]^+^ (D) and 172.0 [C_6_H_4_O_6_]^+^ (E) could correspond to *myo*-inositol after the loss of three and four hydrogen atoms (dehydrogenation) from hydroxyl groups, respectively. In the spectrum before and after electrochemical analysis, the fragment at *m*/*z* = 193.0 (C) can be observed, which may correspond to the addition of a methyl group to the hydroxyl group similarly to L-*chiro*-inositol (Figure 3a–c). Figure 3c shows the ions at *m*/*z* 199.0 (F) and 212.1 (G), which may correspond to the exchange of the methyl group to sodium [C_7_H_12_O_6_ − CH_4_ + Na]^+^, the addition of two oxygen atoms, and the loss of sodium [C_6_H_12_O_6_ − Na + O_2_]^+^, respectively. The ion at *m*/*z* 203.0 [C_6_H_12_NaO_6_]^+^ (H) may correspond to the addition of sodium and the loss of carbon and hydrogen atoms from ion C. The signals at *m*/*z* 219.9 (I), 228.0 (J), 230.9 (K), and 242.2 (L) may correspond to [C_6_H_12_NaO_7_]^+^ (addition of one hydroxyl group, 17 u), [C_6_H_12_O_9_]^+^ (addition of 2 hydroxyl groups, 34 u, and loss of one sodium atom, 23 u), [C_6_H_14_O_9_]^+^ (addition of 2 hydrogen atoms, 2 u) and [C_7_H_14_O_9_]^+^ (addition of methyl group, 12 u), respectively. Ions at *m*/*z* 288.2 [C_13_H_20_O_7_]^+^ (M) and 284.3 [C_13_H_16_O_7_]^+^ (N) may correspond to the addition of hydroxymethyl-phenol and the loss of four hydrogen atoms (dehydrogenation), respectively. Ions observed at *m*/*z* 366.9 (O), 339.0 (P), and 325.2 (R) may be obtained by the addition of C_5_HO_7_ (173 u), the loss of 2 × O, and the addition of 5 × H (27 u) and loss of the CH_2_ group (14 u) from the ring, respectively (Figure 3a). The fragmentation pathway of *myo*-inositol S→T→U→V looks similar to that of L-*chiro*-inositol (Figure 2a and Figure 3a). The ion at *m*/*z* 328.9 (W) could be derived from the ion at *m*/*z* 346.9 (V) by the loss of sodium and the addition of five hydrogen atoms. Moreover, the ion at *m*/*z* 299.0 [C_13_H_15_O_8_]^+^ (X) could be originated from the ion at *m*/*z* 328.9 [C_13_H_12_O_10_]^+^ and was characteristic for *myo*-inositol after the electrochemical analysis. 

#### 2.2.4. D-Glucose

The saccharides can be ionized by the MALDI technique in positive and negative ionization modes. In the positive mode, the fragmentation process was preferentially toward ring–ring cleavage. Furthermore, in the negative mode, the cross-ring cleavage mechanism of fragmentation was dominant [29]. In the case of positive ionization, MALDI-TOF-MS mass spectra for sugars were characterized by similar signals to those of cyclitols. Therefore, mass spectra were present in the negative ionization mode for sugars to show a different fragmentation mechanism. Post-source decay (PSD) spectra of molecular ion at *m*/*z* 179.0 (B) of D-glucose are shown in Figure 4a–c. The ion detected at *m*/*z* 114.6 (C) may correspond to tetrahydropyran with two carbonyl groups [C_5_H_6_O_3_]^−^. The fragment at *m*/*z* 225.0 (D) could be formed by the addition of a carboxyl group (46 u). Ions equivalent to *m*/*z* 261.6 (E) and 247.7 (F) correspond to the addition of a cyclohexane ring and the loss of CH_2_ (cleavage of a cyclohexane ring) (Figure 4a). A dimer of glucose can be observed at *m*/*z* 343.3 [C_12_H_23_O_11_]^−^ (G). The fragment ion at *m*/*z* 345.3 (L) could be formed from the ion at *m*/*z* 343.3 by the addition of hydrogen (hydrogenation). The fragment ion at *m*/*z* 327.3 (H) could be the result also from the ion at *m*/*z* 343.3 by loss of a hydroxyl group. The ion at *m*/*z* 321.4 [C_12_H_17_O_10_]^−^ (M) could be formed by the loss of six hydrogen atoms from the ion G, and probably in this case, the ring closes between the free hydroxyl groups (Figure 4a). The ions detected at *m*/*z* 307.4 (I), 289.5 (J), and 273.6 (K) can be ascribed to [C_12_H_19_O_9_]^−^ (loss of a hydroxyl group and three hydrogen atoms, 20 u), [C_12_H_17_O_8_]^−^ (loss of water molecules, 18 u), [C_12_H_17_O_7_]^−^ (loss of an oxygen atom, 16 u), respectively. The ion at *m*/*z* 315.4 [C_10_H_19_O_11_]^−^ (N) could be derived from the molecular ion [C_6_H_11_O_6_]^−^ by the addition of a “glucose ring” (Figure 4a). The loss of hydrogen from the ion at *m*/*z* 315.4 may produce the ion at *m*/*z* 314.7 [C_10_H_18_O_11_]^−^ (O).

#### 2.2.5. D-Fructose

MALDI-TOF-MS spectra of D-fructose are shown in Figure 5a–c. D-fructose belongs to monohexoses and Figure 5b presents the deprotonated molecule [M − H]^−^ at *m*/*z* 179.4 (B). The signal at *m*/*z* 170.2 (C) may correspond to deprotonated D-fructose [C_6_H_2_O_6_]^−^. The signal at *m*/*z* 114.7 (D) could be created from the molecular ion B by the loss of three hydroxyl groups and loss of a CH_2_ group. The ion at *m*/*z* 289.6 (E) can be attributed to the addition of a C_5_H_2_O_3_ (110 u) molecule to a molecular ion. The ion at *m*/*z* 267.7 [C_9_H_15_O_9_]^−^ (F) may be formed due to the cross-ring cleavages and the loss of a carbon atom (Figure 5a). The signal at *m*/*z* 251.7 [C_9_H_15_O_9_ − O]^−^ (G) could be formed due to the loss of an oxygen atom from the ion at *m*/*z* 267.7. The sodium adduct with two molecules of D-fructose followed by the loss of two hydrogen atoms may correspond to the ion at *m*/*z* 361.4 [C_12_H_18_NaO_11_]^−^ (H). The ion observed at *m*/*z* 377.3 [C_12_H_18_NaO_11_ + O]^−^ (I) can be formed by the addition of an oxygen atom. The loss of one hydrogen atom (dehydrogenation) from the ion at *m*/*z* 361.485 may form the ion at *m*/*z* 360.3 [C_12_H_17_NaO_11_]^−^ (J). In addition, the loss of an oxygen atom from the ion at *m*/*z* 361.4 could create the ion at *m*/*z* 345.4 [C_12_H_18_NaO_10_]^−^ (K). The fragment formation at *m*/*z* 329.4 (L) with low abundance can correspond to [C_12_H_18_NaO_9_]^−^ (loss of an oxygen atom from the ion at *m*/*z* 345.4). The addition of four hydrogen atoms (hydrogenation) and cross-ring cleavages from the ion at *m*/*z* 329.4 can produce the fragment at *m*/*z* 321.5 [C_12_H_18_NaO_9_ − C + 4H]^−^ (M). The signals detected at 315.5 (N) and 307.6 (O) can be ascribed to [C_11_H_22_NaO_9_ − 5H]^−^ (the loss of five hydrogen atoms and the formation of a radical anion) and [C_11_H_16_NaO_9_ − C + 4H]^−^ (the loss of one carbon atom and addition of four hydrogen atoms), respectively (Figure 5a).

#### 2.2.6. D-Galactose

The ion at *m*/*z* 179.2 (B) was detected in the MS spectra for D-galactose [M − H]^−^ (Figure 6a–c). The signals at *m*/*z* 170.1 (C) and 114.6 (D) may correspond to [C_6_H_2_O_6_]^−^ (loss of eight hydrogen atoms and formation of a radical anion) and [C_5_H_6_O_3_]^−^ (loss of three hydroxyl groups and loss of a CH_2_ group). The fragments at *m*/*z* 315.4 [C_11_H_23_O_10_]^−^ (E) may correspond to [M − H + C_5_H_11_O_5_]^−^ from a molecular ion. The loss of 3 × OH and loss of a CH group from the ion at *m*/*z* 315.4 could create the ion at *m*/*z* 251.7 (F). The signal at *m*/*z* 233.8 [C_9_H_13_O_7_]^−^ (G) can be obtained from the ion at *m*/*z* 251.7 [C_10_H_19_O_7_]^−^ by the loss of CH_6_ (18 u). The ion observed at *m*/*z* 361.1 [C_12_H_18_NaO_11_]^−^ (N) could be formed by the addition of one molecule of D-galactose and sodium and the loss of two hydrogen atoms (Figure 6a). The addition of a hydroxyl group (17 u) to the ion observed at *m*/*z* 361.1 could create the ion at *m*/*z* 377.2 [C_12_H_18_NaO_11_ + OH]^−^ (P). Moreover, the loss of six hydrogen atoms from the ion at *m*/*z* 377.2 could produce the ion at *m*/*z* 371.3 [C_12_H_18_NaO_12_ − 6H]^−^ (R). The small intense signal at *m*/*z* 360.3 (O) could be formed by the dehydrogenation process from the ion [C_12_H_18_NaO_11_]^−^ (Figure 6c). The ions at *m*/*z* 345.3 (H), 329.3 (I), 307.4 (K), 289.4 (L), and 273.5 (M) probably corresponding to [C_12_H_25_O_11_]^−^ (addition of one molecule of D-galactose and four hydrogen atoms with the cross-ring cleavages), [C_12_H_25_O_10_]^−^ (loss of one hydroxyl group, 17 u), [C_12_H_19_O_9_]^−^ (loss of one hydroxyl group, 17 u, and five hydrogen atoms, 5 u), [C_12_H_17_O_8_]^−^ (loss of one water molecule, 18 u) and [C_12_H_17_O_7_]^−^ (loss of an oxygen atom, 16 u), respectively. Additionally, the fragment at *m*/*z* 321.3 [C_12_H_17_O_10_]^−^ (J) could be generated from the ion at *m*/*z* 329.3 by the loss of eight hydrogen atoms (dehydrogenation). 

In general, analysis of the obtained mass spectra (Figure 1, Figure 2, Figure 3, Figure 4, Figure 5 and Figure 6) has shown that some signals appeared only after the electrochemical process (Figure 1c, Figure 2c, Figure 3c, Figure 4c, Figure 5c, Figure 6c), and some signals had different intensities before and after the electrochemical process. This can be evidence of the occurrence of possible metabolites of the analyzed compounds. Regarding cyclitols, Figure 1b,c have shown that many signals (denoted as E, G, I, J, K, and R) occurred only after the electrochemical process. The ion at *m*/*z* 265.9 disappeared after the electrochemical oxidation. In addition, the fragments T and S (denoted in red color) had higher intensity before the electrochemical analysis (Figure 1b,c). The electrochemical oxidation for chiro-inositol also seemed to generate many signals since they occurred only after the EC fragmentation. The signals E, N, M, R, S, T, U, V, and W were found to be characteristic for chiro-inositol and thus can be indicators of the simulated bioconversion process. In the spectrum in Figure 2b, it can observed that ions at *m*/*z* 244.2 and 256.2 disappeared. In case of myo-inositol, characteristic signals also appeared as well as in the case of L-*chiro*-inositol (Figure 3b,c). Moreover, *chiro*-inositol and *myo*-inositol are supposed to have higher affinity for the formation of DHB and HCCA matrix adducts compared to D-pinitol [30]. Considering saccharides before and after EC assay, fewer products of electrochemical conversion are observed. For D-glucose and D-galactose, only five signals appeared after the electrochemical process (Figure 4c and Figure 6c). In the case of D-fructose (Figure 5c), only one signal can be observed, which appears in the spectrum after the electrochemical process. Moreover, the saccharides have a high ability to form dimers (disaccharides).

Table 1 presents the fragments of sugars and cyclitols that were suggested to be formed during the electrochemical oxidation and detected by MALDI-TOF-MS. Fragments arising from reactions such as hydrogenation/dehydrogenation and hydroxylation can be considered as possible metabolites, since these are mainly expected metabolic reactions of phase I oxidative cell metabolism. Cross-ring cleavage was also suggested to occur for saccharides. Fragments coming from the association of molecules with sodium (metal adducts), matrix compounds, or units that belong to the core of the matrix’s structure (matrix adducts) cannot be considered as potential metabolites that could be produced endogenously. However, such adducts may be valuable for compound identification using the present technique; additionally, metabolic reactions that might occur with the target analyte still can be observed. Therefore, the mentioned fragments were also included in Table 1. Compounds marked with an asterisk have a higher probability to be formed endogenously. Other fragments may refer to less probable candidate metabolites, since MALDI ionization adducts may be formed due to the lower stability of the parent molecule. Once the proposed approach consists of an *off-line* mode, only stable analytes could be analyzed. Nevertheless, the formation of adducts during ionization indicates that these are thermodynamically less stable forms in a gas phase, unlikely acting as effective bioactive compounds in the organism because are prone to further rearrangement. Thus, it is important to consider that candidate metabolites created in the electrochemical cell were identified in this study by their ionization in a gas phase. Subsequently, the study is addressed to model potential metabolic reactions that may occur in phase I of cell metabolism. Enzymatic reactions also should be taken into account to propose metabolites for studied analytes similarly to the application of state-of-the-art *on-line* EC-LC-MS approach. 

The application of MALDI-TOF-MS for the analysis of sugar and cyclitol transformations is a new approach in the identification of candidate metabolites. Additionally, little is known regarding the elucidation of the fragmentation pathways of these products or the determination of such compounds in real samples. Only a few papers reporting on the application of the MALDI technique for the determination of sugars and cyclitols can be found in the literature [30,31,32,33,34,35].

### 2.3. Data Analysis Approaches 

Network analysis (NA) depicted relationships (edges) among groups of studied compounds (nodes), allowing a visualization of the ions (generated MS fragments) concurrent between sugars and cyclitols and their correspondent variants, which were submitted to the electrochemical method (Figure 7). Overlapping ions refer to ions connected to more than one node, while ions specifically incident in the MS spectrum of a given analyte are represented by loose edges coming from a corresponding node. Each of the species (colored blocks) emit lines, which can be connected to a second compound, demonstrating that they have that feature in common. For example, in Figure 7a, *m*/*z* 268 is present in the MS spectra of D-pinitol, D-galactose, *myo*-inositol, and also the D-fructose variant after EC. On the other hand, *m*/*z* 228 is a fragment that is exclusively found in D-pinitol, once the edge representing this fragment is emitted by the D-pinitol node and is not connected to any other node. In this way, fragments situated in the center of the net are shared between most of the nodes surrounding it; thus, they are shared between many species. Those located at the periphery of the net are more exclusive fragments that are less recurrent in the MS spectra of analytes. Concurrent fragments were evidenced in the case of all used sets of analysis parameters, suggesting a relevant relationship between the fragmentation patterns of the different investigated compounds. This may be due to the similarities in their original structures and in the processes of adduct formation. Regarding the DHB matrix, the positive ionization mode yielded more correlated spectra between the original and electrochemically processed analytes, whereas for the HCCA matrix, the opposite is observed. Concerning the observation of distinctive ions with greater intensity, these were hindered in the case of EC-treated analytes when the DHB matrix was used. Such an aspect may hamper the differentiation of such compounds in impurified samples.

The relative distance between the nodes also indicates their level of correlation based on the number of interconnected variables. For example, more intricate networks are formed by using DHB in the positive mode and HCCA in the negative mode. This shows that in these conditions, less unique MS spectra are obtained for the investigated species in terms of incident ions. Additionally, different patterns of correlation are observed, depending on the used analytical parameters. However, there is a more prominent grouping between species submitted to the electrochemical assay, indicating that the species obtained from this process share more fragmentation products with considerable intensity.

Considering the analysis in the positive mode and using the DHB matrix, the ions shared between saccharides and cyclitols—original and artificially metabolized—were *m*/*z* 377 and *m*/*z* 393, with *m*/*z* 193 being absent solely in the D-glucose spectrum. The fragment *m*/*z* 377 corresponds to the association of a sodium adduct of methylated [M + H]^+^ ion with a DHB unit (in the case of D-pinitol, it corresponds to the demethylated sodium adduct of the dimerized form); *m*/*z* 393 can be addressed as a precursor of the later fragment, due to an oxygen loss in the structure. The ion *m*/*z* 193 may be ascribed to [C_7_H_12_O_6_ + H] ^+^ or [C_6_H_8_O_7_ + H] ^+^ fragments, the first one preferably generated by cyclitols and the second one by D-fructose and D-galactose. 

The fragments *m*/*z* 361, 338, 376, and 339 were coincident among all species after the electrochemical treatment. The ion *m*/*z* 343 can be formed by dimerization; in D-pinitol, it refers to the dimer after the loss of 2H and one methyl group; *m*/*z* 361 can be related to the latter after the gain of one hydroxyl group and one proton (or simply, in the case of L-*chiro*-inositol and *myo*-inositol, to [2M + H]^+^); *m*/*z* 376 can be related to the gain of one oxygen and one hydroxyl molecule; *m*/*z* 339 can be related to the abstraction of 4H (in D-pinitol and *myo*-inositol, *m*/*z* 339 may also be generated from [M + H]^+^ by the combined associations: + M + Na + DHB unit). Finally, *m*/*z* 338 can be produced by the subtraction of oxygen in ion *m*/*z* 354 in cyclitols and from the subtraction of 4H in the ion *m*/*z* 342 in sugars (generated by [M+H]^+^ combined with one more molecule).

Ions such as *m*/*z* 230, 233, and 288 were coincident solely in both species before electrochemical processing. The fragment *m*/*z* 203 can be connected with the ion [M + Na]^+^ (in cyclitols, it can be obtained by the scission of [2M + H]^+^ and subsequent combination with sodium and abstraction of a hydrogen atom), which in combination with an ethyl group yields *m*/*z* 230. The same process having as its precursor [M + H + Na]^+^ may lead to *m*/*z* 233. The ion *m*/*z* 288 may refer to the association of a cyclitol molecular ion (or its directed derivative -H, for D-pinitol) together with a benzaldehyde unit. In saccharides, the association of [M + H]^+^ with the moiety C_3_H_8_O_3_ can lead to *m*/*z* 273 in the spectrum, in which methylation can give rise to *m*/*z* 288.

Using the HCCA matrix, in the positive mode, the ions *m*/*z* 192 and *m*/*z* 189 appear to be shared among both original and treated analytes. [M + H]^+^(or scission product of cyclitol core in [2M + H]^+^) suffers water loss and if combined with the formyl group, it produces the fragment *m*/*z* 192. If associated with the loss of a CN group from the matrix, the ion *m*/*z* 189 is produced. Ions *m*/*z* 284 and *m*/*z* 289 were common for all species subjected to the electrochemical process: the first can be obtained by the addition of moiety C_4_H_7_O_3_ to the molecular ion and the second can be obtained by association of the latter with the benzyl alcohol portion of the 4-HCCA molecule. Still regarding this analysis method, *m*/*z* 288 again appears recurrent in all analytes in their original form. In negative mode analysis, any characteristic ions were found regarding the groups of studied chemical species.

HCA (hierarchical cluster analysis) (Figure 8a,b) provides information regarding sample grouping with its basis on the calculated dissimilarity coefficient. The similarity between two MS spectra is measured according to a parameter, in this case, the Spearman coefficient. The level of correspondence between a pair of profiles that are being compared is related to height of the dendrogram: the shorter the height, the greater the similarity. In this way, in HCA, the assays are spatially organized according to their level of congruence with each other. Correspondence between the obtained MS spectra did not enable a clear discrimination between saccharides and cyclitols species once these did not form an isolated cluster for each type of ionization mode. However, compounds after EC appeared grouped together, indicating that analytes subjected to electrochemical assay generated more coincident fragments, and thus, coincident possible metabolites. Figure 8c presents a chart with ions incident solely in the MS spectrum of one of the analytes, showing that distinctive ions were not detected for most of the performed assays, as expected due to the structural similarity of analytes. The formation of an ion at *m*/*z* 152 [C_5_H_11_O_5_ + H]^+^ was favored for L-*chiro*-inositol, it consists of a molecular ion after dehydrogenation and decarboxylation. The ion at *m*/*z* 207 may refer to the gain of –CH and the loss of a proton in the region of D-pinitol’s methyl radical, leading to an epoxy structure, while *m*/*z* 228 can be obtained after the loss of a methyl group, followed by oxidation (the gain of two atoms of oxygen and one hydroxyl group). For *myo*-inositol, a signal at *m*/*z* 325 appears as a specific fragment that was probably generated by a loss of water followed by the loss of CH_2_O in a molecule dimer. D-glucose has associated the ion at *m*/*z* 256, which may arise from the combination of the original structure with C_3_H_7_O_2_. The methylation and dehydrogenation of a glucose dimer can result in the fragment at *m*/*z* 356; at the same time, *m*/*z* 330 in the D-fructose MS spectrum can be a product of hydroxymethyl group loss followed by hydroxyl gain. D-galactose ion *m*/*z* 276 can be generated by the loss of two molecules of water and one –CH_2_O group in the structure of the dimer (*m*/*z* 343). Species that were electrochemically transformed also presented distinguishable fragments. The ion at *m*/*z* 384 can be a product of L-chiro-inositol dimerization with posterior dehydrogenation and the incorporation of sodium. The fragment *m*/*z* 274 from D-pinitol may result from *m*/*z* 387 [2M − H]^+^, followed by the loss of two –CH_2_O groups and three molecules of water. Transformed *myo*-inositol is characterized by the *m*/*z* 285 ion, which may refer to [2M − H]^+^ after the loss of four molecules of water. In negative mode, transformed L-chiro-inositol was associated to the formation of a signal at *m*/*z* 188, which may correspond to [C_7_H_8_O_6_]^−^ after methylation and successive dehydrogenations. Although investigated analytes are structurally similar and display corresponding molecular weights, the verification of ions specific for some of the studied species suggests that some mechanisms related to molecule rearrangement and fragmentation may be favored to the detriment of others in circumstances of factors such as the steric effect.

Figure 9 presents a scatter plot of values of fold change in ion intensity after compounds were electrochemically treated. All the represented ions are those that suffered statistically relevant changes in their intensities in comparison to the EC-treated form. The increasing size of plotted variables indicates their increasing level of significance (expressed in terms of common logarithm of *p* value). Considering this, bigger dots refer to fragments associated to smaller p-values. Colors of features represent four main responses in the spectra obtained after modification of the products: ions that have become absent after the EC process, those that have become incident, and those that increased or decreased their intensities after EC treatment. The graphs allow to observe that, in positive mode, the DHB matrix provided spectra with a greater number of discriminating ions. In negative mode, only a few significant changes between the compounds’ spectra before and after treatment were observed. This aspect is probably related to a greater dissimilarity occurring among the spectra of sugars and cyclitols in negative mode analysis. The existence of such discriminating variables denotes the resemblance of MS profiles belonging to the two investigated groups of compounds, a factor that can hinder a proper differentiation between them. However, these features may serve as indicative species that are useful in the monitoring of the metabolization process of the studied compounds. Together with previously presented evidence, the compounds subject to the electrochemistry step tend to present MS spectra containing a greater variety of ions addressed to the total or partial combination of molecular ions with moieties of itself or with matrix compounds. In general, the addressed fragments involve dimerization, alterations, and sodium adduct formation in the dimer structure as well as combination with DHB and 4-HCCA molecules. This suggests that the transformed varieties of analytes tend to be more prone to recombination, due to their increased reactivity. After processing, the MS spectra similarity increases and the number of diverse fragments decreases, pointing to an allowed interconversion of these species into coincident structures, which can refer to ions marked in Figure 9 with “incident” behavior after EC assay. Ions that had their intensities significantly decreased are related to fragments coming from minor transformations in the molecular ion unit. Those related to punctual chemical alterations of molecular ions, or from the fragmentation of original structure itself, tend to appear “absent” after EC assay. Such approach provides an overview of the significance of possible metabolites. The conversion of bioactive compounds as a part of metabolism by cells is important to consider for biological activity studies and thus, new approaches for the simulation of cell metabolism should be proposed. The developed *off-line* EC-MALDI-TOF-MS approach can find its application in the metabolomics studies for the identification of candidate metabolites of potential drugs. In addition, the approach can be interesting for researchers in the field of natural bioactive compounds since, as it was mentioned above, the identification of metabolites can be valuable for the assessment of biological activity.

## 3. Materials and Methods 

### 3.1. Chemicals and Reagents

D-pinitol (95% mass), L-*chiro*-inositol (95% mass), *myo*-inositol (≥98% mass), D-glucose, D-fructose, and D-galactose were purchased from Sigma-Aldrich (Steinheim, Germany). Water, acetonitrile, methanol (all solvents were of LC-MS grade purity), trifluoroacetic acid (≥99% purity), and ammonium acetate (≥99% purity) were purchased from Sigma-Aldrich (Steinheim, Germany). MALDI matrices such as 2,5-dihydroxybenzoic acid (DHB), α-cyano-4-hydroxycinnamic acid (HCCA) as well as cesium triiodide for mass calibration were purchased from Sigma-Aldrich (Steinheim, Germany). Water was obtained with a Milli-Q RG apparatus by Millipore (Millipore Intertech, Bedford, MA, USA).

### 3.2. Sample Preparation

Stock solutions of standards were prepared by dissolving standards of cyclitols and saccharides in water (Milli-Q). Solutions for subjection to electrochemical conversion were prepared at neutral pH by dissolving 50 µL of standard in 5 mL of 10 mM ammonium acetate and 2 mL of acetonitrile to a final concentration of 10 µg/mL. Control samples that were not subjected to EC conversion were prepared by the same procedure. Two fractions of each compound after EC conversion were collected to 2-mL Eppendorf tubes (the first fraction was collected during the first 10 min and the second was collected during the next 7 min). After collection of the fractions, the solvent was evaporated and frozen in −80 °C. Prior to analysis, samples were re-dissolved in 50 µL of methanol–water (1:1) mixture.

### 3.3. Instrumentation

The ROXY™EC system (Antec, Zoeterwoude, The Netherlands) was used for the electrochemical conversion of analytes. The system consisted of a potentiostat equipped with a ReactorCell™ with a reference electrode HyREF™ (Pd/H_2_) and working electrode (boron-doped diamond electrode, BDD), an infusion pump (Harvard, Holliston, MA, USA), and all necessary capillary tubes. The Dialogue^TM^ version 2.02.145 software (Antec, Zoeterwoude, The Netherlands) was used. The BDD electrode consisted of an ultra-thin layer deposited on top of a silicon substrate in a potential range from 0 to 3000 mV. Additionally, an electrochemical three-electrode arrangement was composed of a Pd counter electrode and a HyREF (Pd/H_2_) reference electrode. All working solutions of cyclitols were freshly prepared from stock solutions at final concentrations of 10 µg/mL. Working solutions of cyclitols and saccharides were prepared using 10 mM ammonium acetate (pH 7.4) with the addition of 2 mL of acetonitrile and injected to the system using a 1-mL syringe (Hamilton, Reno, NV, USA). The analysis was performed using a flow rate of 10 μL/min and 37 °C oven temperature. All the fractions and controlled samples were transferred to 2-mL Eppendorf tubes and analyzed by MALDI-TOF-MS (Bruker Daltonics, Bremen, Germany). 

### 3.4. MALDI-TOF-MS

The samples of cyclitols and saccharides were analyzed by an ultra-fleXtreme MALDI-TOF-MS instrument (Bruker Daltonics, Bremen, Germany). The instrument is equipped with a modified neodymium-doped yttrium aluminum garnet (Nd:YAG) laser (Smart beam II™) operating at the wavelength of 355 nm and frequency of 2 kHz, which was used for all measurements. The spectra were analyzed in a reflective positive and negative ionization mode in the 60–1600 *m*/*z* range at 80% of laser power and global attenuator at 50%. Fragment spectra were determined using the LIFT mode in the *m*/*z* range of 50–1000. All mass spectra were acquired and processed using software such as Flex Control and Flex Analysis, respectively (Bruker Daltonics, Bremen, Germany).

The matrices such as α-cyano-4-hydroxycinnamic acid (HCCA) and 2,5-dihydroxybenzoic acid (DHB) at a concentration of 10 mg/mL were prepared by dissolving 10 mg of HCCA and DHB in 1 mL of standard solution (30% acetonitrile 70% H_2_O and 0.1% trifluoroacetic acid. The mixtures of 1 µL of each sample and 1 µL of matrix solutions were applied to the spot on a MALDI-TOF-MS MTP AnchorChip 384 plate, in triplicate. All mass spectra were calibrated by using the cesium triiodide-cluster (CsI_3_); 10 mg of CsI_3_ was dissolved in 1 mL of mixtures of methanol–DHB (1:1).

### 3.5. Exploratory Data Analysis 

The following methods were conducted in R environment, using RStudio console v. 1.1.463 (RStudio, Boston, MA, USA) and employing the packages “gplots”, “sna”, and “ggplot2”. NA aimed to represent coincident ions among the spectra of the investigated compounds. HCA using the Spearman rank correlation method was conducted in order to observe the grouping of performed assays according to the distribution of MS ions. For NA, data concerning only ions with at least 1% of the total spectral abundance were considered, and the dataset was converted into binary entries. For HCA input, Z-score normalization of the data was carried out. Finally, weighted scatter plots were intended to highlight the main significant trends presented by MS ions when submitted to the electrochemical method. These results were expressed in terms of binary logarithm of the fold change in ion intensity. Using IBM SPSS Statistics v. 24 (IBM, Armonk, NY, USA), the Mann–Whitney U test was performed, aiming to point out discriminant features between the group of analytes in original form and when submitted to the electrochemical process; a *p* value < 0.05 was considered as the significance criterion. All prepared datasets comprehended ions within the range of 150 to 400 *m*/*z*. Matrix spectra (“blanks”) were subtracted from analyses, and the average of intensity values coming from fractions 1 and 2 were used. Exploratory data analysis encompassed the general fragments generated by MALDI-TOF-MS analysis, obeying the aforementioned thresholds of intensity and statistical significance.

## 4. Conclusions

The study used electrochemical analysis with MALDI-TOF-MS detection for the first time to identify candidate metabolites of biologically active compounds. Using the *off-line* EC-MALDI-TOF-MS method, it was possible to identify and characterize the EC-generated profile of cyclitols (pinitol—*m*/*z* 346.9, 354.9, 361.9, 370.9, chiro-inositol—*m*/*z* 317.0, 331.0, 376.9, *myo*-inositol—*m*/*z* 172.0, 199.0, 212.1, 284.3, 299.0, 328.9, 339.0, 346.9, 360.9) and sugars (glucose—*m*/*z* 225.0, 315.4, 321.4, 345.3, fructose—*m*/*z* 329.4, galactose—*m*/*z* 233.8, 251.7, 321.3, 360.3). In addition, the identified products were characteristic for the sugars and cyclitols tested. The use of electrochemistry is a new approach for the simulation and detection of possible bioconversion products using electrochemical reactions. In addition, the use of statistical analysis showed differences/similarities between MS spectra acquired for compounds before and after the electrochemical process for the two tested matrices. EC assay products provided less distinguishable MS spectra and were mainly characterized by the combination of analytes with more complex structures. The applied method proved to be adequate for the identification and assessment of the analytes and their derived species to model potential metabolic reactions of phase I oxidative cell metabolism for cyclitols and saccharides. Candidate metabolites of the studied substances were indicated, and these results can be introduced in further studies regarding the bioactivity of these compounds, influencing parameters, and in the design of possible pharmacological applications.

## Figures and Tables

**Figure 1 ijms-21-05265-f001:**
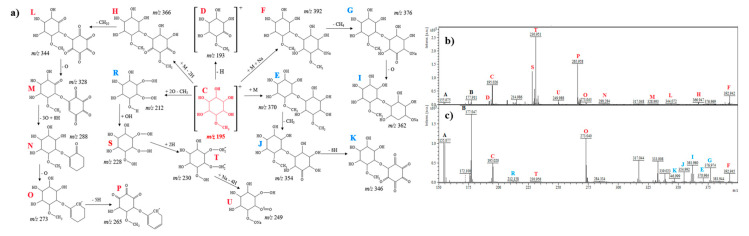
(**a**) Fragmentation pathways of D-pinitol; (**b**) MS spectrum before electrochemical process, (**c**) MS spectrum after electrochemical process.

**Figure 2 ijms-21-05265-f002:**
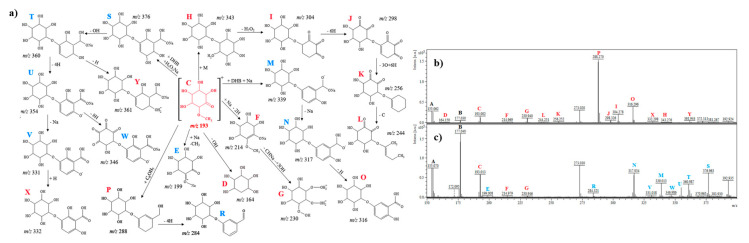
(**a**) Fragmentation pathways of L-*chiro*-inositol; (**b**) MS spectrum before electrochemical process, (**c**) MS spectrum after electrochemical process.

**Figure 3 ijms-21-05265-f003:**
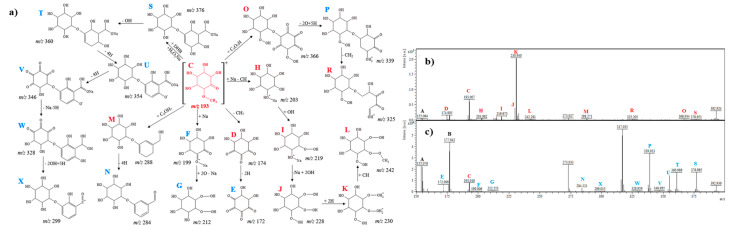
(**a**) Fragmentation pathways of myo-inositol; (**b**) MS spectrum before electrochemical process, (**c**) MS spectrum after electrochemical process.

**Figure 4 ijms-21-05265-f004:**
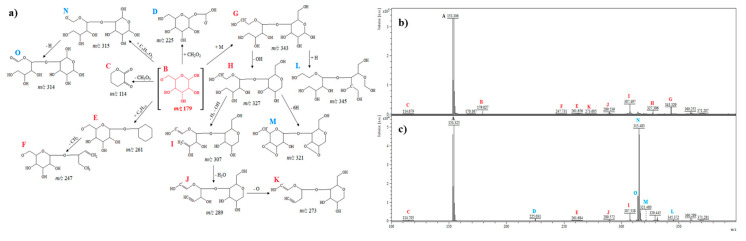
(**a**) Fragmentation pathways of D-glucose; (**b**) MS spectrum before electrochemical process, (**c**) MS spectrum after electrochemical process.

**Figure 5 ijms-21-05265-f005:**
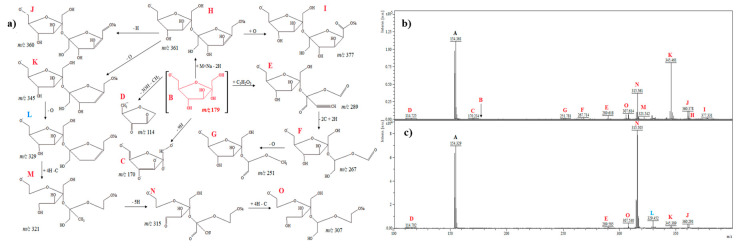
(**a**) Fragmentation pathways of D-fructose; (**b**) MS spectrum before electrochemical process, (**c**) MS spectrum after electrochemical process.

**Figure 6 ijms-21-05265-f006:**
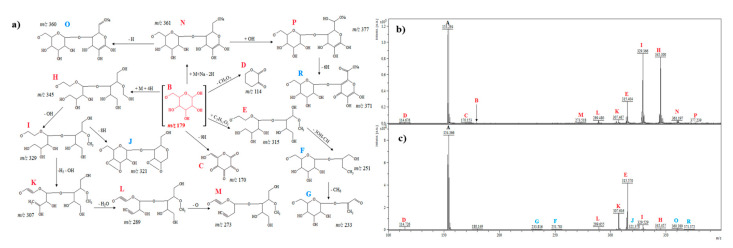
(**a**) Fragmentation pathways of D-galactose; (**b**) MS spectrum before electrochemical process, (**c**) MS spectrum after electrochemical process.

**Figure 7 ijms-21-05265-f007:**
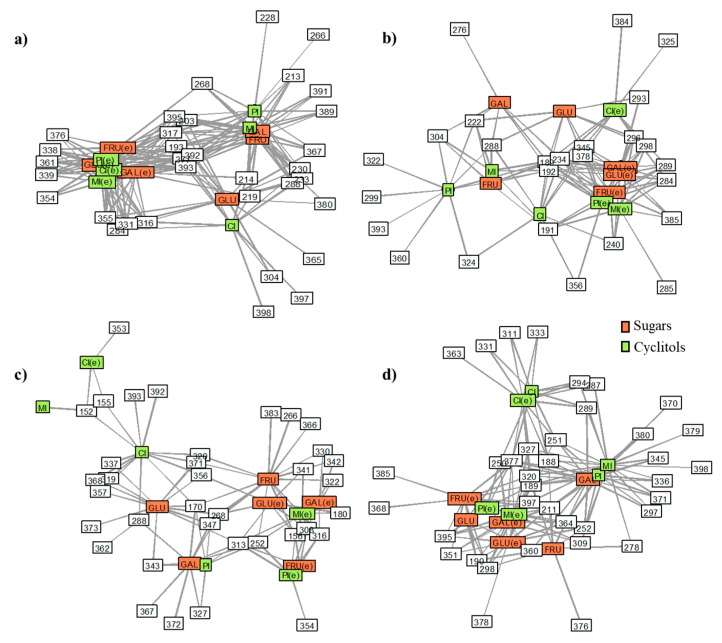
Networks built based on MS spectra obtained using (**a**) 2,5-dihydroxybenzoic acid (DHB) and (**b**) α-cyano-4-hydroxycinnamic acid (HCCA) matrices, in positive ionization mode; (**c**) DHB and (**d**) HCCA matrices, in negative ionization mode. Colored nodes: compounds, white rectangles: MS ions, e—compounds subjected to the electrochemical process, FRU: D-fructose, GAL: D-galactose, GLU: D-glucose, PI: D-pinitol, CI: L-*chiro*-inositol, and MI: *myo*-inositol.

**Figure 8 ijms-21-05265-f008:**
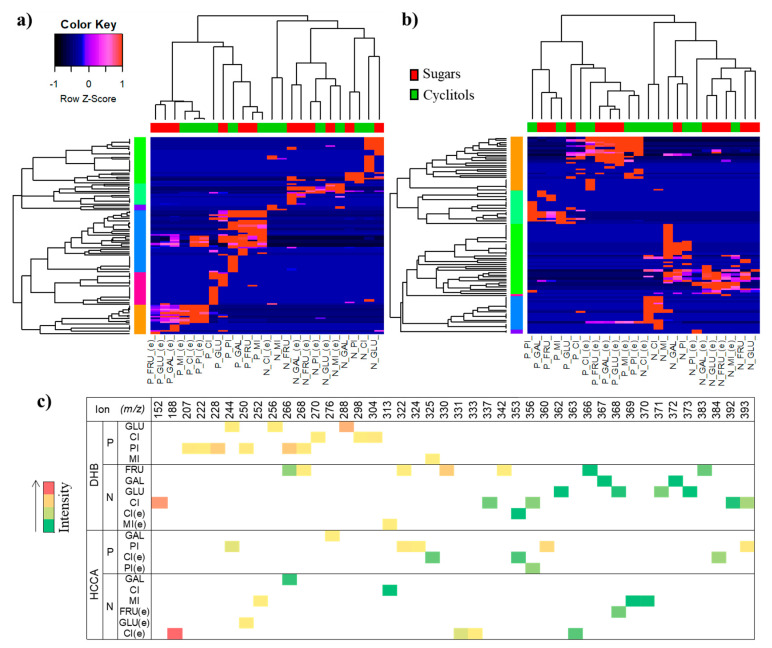
Heatmaps associated to hierarchical cluster analysis (HCA), which were generated for analysis using (**a**) DHB and (**b**) HCCA matrix; (**c**) chart showing ions specifically incident in the spectrum of one of the analytes. P: positive mode, N: negative mode, e—compounds subject to electrochemical process, FRU: D-fructose, GAL: D-galactose, GLU: D-glucose, PI: D-pinitol, CI: L-chiro-inositol, MI: *myo*-inositol.

**Figure 9 ijms-21-05265-f009:**
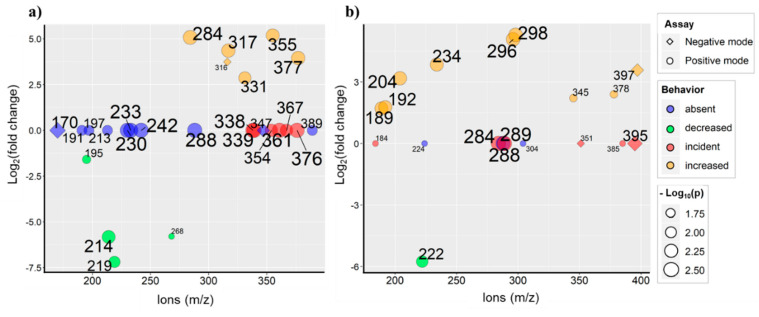
Scatter plots weighted according to the significance of ions (-log_10_ (p)), referring to a comparison between MS spectra obtained before and after the electrochemical process, using (**a**) DHB and (**b**) the HCCA matrix.

**Table 1 ijms-21-05265-t001:** Fragments electrochemically generated for cyclitols and sugars, characterized by different *m*/*z* values and the proposed involved reaction (* = fragments referring to candidate metabolites once these were assumed to be possibly produced in a biological system).

Compounds	Fragment	*m*/*z*	Ionization	Proposed Reaction
**D-pinitol**	C_7_H_6_O_5_	172.1	+	OH − 7H loss (dehydroxylation/dehydrogenation)
	* C_6_H_12_O_8_	212.1	+	CH_3_ loss − 2 OH gain (hydroxylation)
	C_13_H_16_O_7_	284.3	+	CH_3_ loss − Benzaldehyde gain
	C_13_H_17_O_10_	333.0	+	Cyclohexane + H_2_O + 3O gain (hydration/reduction)
	C_13_H_16_NaO_9_	339.0	+	CH_3_ loss − DHB + Na gain (adduct with matrix and metal)
	C_13_H_14_O_11_	346.9	+	Cyclohexane + 5O gain (reduction)
	C_13_H_22_O_11_	354.9	+	Cyclohexane + 4OH + O gain (hydroxylation/reduction)
	C_13_H_23_NaO_10_	361.9	+	D-pinitol + Na gain − OH − CH_3_ loss (dimerization/adduct with metal/dehydroxylation)
	C_14_H_26_O_11_	370.9	+	D-pinitol gain (dimerization)
	C_15_H_27_O_11_	383.9	+	D-pinitol + =CH_2_ gain (dimerization)
	C_13_H_22_NaO_10_	360.3	−	D-pinitol + Na gain − OH − CH_3_ loss (dimerization/adduct with metal/dehydroxylation)
	C_20_H_36_O_12_	467.1	−	D-pinitol + phenol gain (dimerization)
	C_20_H_35_NaO_12_	490.2	−	D-pinitol + phenol + Na gain (dimerization/adduct with metal)
	C_20_H_35_NaO_14_	522.9	−	D-pinitol + phenol + Na + 2OH gain (dimerization/adduct with metal/hydroxylation)
	C_27_H_48_O_18_	660.0	−	2D-pinitol + phenol + OH gain (dimerization/hydroxylation)
**L-chiro-inositol**	C_6_H_4_O_6_	172.0	+	8H loss (dehydrogenation)
	C_6_H_8_NaO_6_	199.0	+	Na gain − 4H loss (adduct with metal/dehydrogenation)
	C_13_H_16_O_7_	284.3	+	Benzaldehyde gain
	C_13_H_17_O_9_	317.0	+	DHB gain (adduct with matrix)
	C_13_H_15_O_10_	331.0	+	DHB + OH gain (adduct with matrix/hydroxylation)
	C_13_H_16_NaO_9_	339.0	+	DHB + Na gain (adduct with matrix and metal)
	C_13_H_7_NaO_10_	346.9	+	DHB + Na + OH gain − 10H loss (adduct with matrix and metal/hydroxylation/ dehydrogenation)
	C_13_H_15_NaO_10_	354.8	+	DHB + Na + OH gain (adduct with matrix and metal/hydroxylation)
	C_13_H_21_NaO_10_	360.9	+	DHB + Na + OH gain + 4H (adduct with matrix and metal/hydroxylation/hydrogenation)
	C_14_H_26_O_11_	370.9	+	L-chiro-inositol + 2CH_3_ gain (dimerization)
	C_13_H_21_NaO_11_	376.9	+	DHB + Na + 2OH gain − 4H (adduct with matrix and metal/hydroxylation/ dehydrogenation)
	C_14_H_23_O_12_	383.9	+	L-chiro-inositol + C_2_H_4_ + =O gain (dimerization/reduction)
	* C_6_H_10_O_6_^∙^	178.8	−	H loss (dehydrogenation)
	C_12_H_15_O_6_	255.9	−	Benzene gain
	C_12_H_15_O_7_	271.9	−	Benzene + OH gain (hydroxylation)
	C_12_H_15_O_10_	319.9	−	Benzene + 4OH gain (hydroxylation)
	C_18_H_31_O_13_	455.9	−	L-chiro-inositol + phenol + OH gain (dimerization/hydroxylation)
	C_24_H_37_O_19_	629.0	−	2L-chiro-inositol + phenol + 2O gain (dimerization/reduction)
	C_24_H_41_O_20_	649.0	−	2L-chiro-inositol + phenol + 3OH gain (dimerization/hydroxylation)
**Myo-inositol**	C_6_H_4_O_6_	172.0	+	8H loss (dehydrogenation)
	C_6_H_8_NaO_6_	199.0	+	Na gain − 4H loss (adduct with metal/dehydrogenation)
	* C_6_H_12_O_8_	212.1	+	2OH gain (hydroxylation)
	C_13_H_16_O_7_	284.3	+	Benzaldehyde gain
	C_13_H_15_O_8_	299.0	+	Benzene + OH + C=O gain (hydroxylation/reduction)
	C_13_H_12_O_10_	328.9	+	DHB + OH gain − 4H loss (hydroxylation/dehydrogenation)
	C_12_H_19_O_11_	339.0	+	Cyclohexane + 2OH + =2O + H_2_O gain (hydroxylation/reduction/hydration)
	C_13_H_7_NaO_10_	346.9	+	DHB + Na + OH gain − 10H loss (adduct with matrix and metal/hydroxylation/ dehydrogenation)
	C_13_H_21_NaO_10_	360.9	+	DHB + Na + OH gain + 4H (adduct with matrix and metal/hydroxylation/ hydrogenation)
	C_12_H_21_O_6_	261.7	−	Cyclohexane gain
	C_12_H_21_O_10_	325.5	−	Cyclohexane + 4OH gain (hydroxylation)
	C_18_H_31_O_13_	455.2	−	Myo-inositol + Cyclohexane + 2OH gain (dimerization/hydroxylation)
	C_18_H_31_O_16_	503.2	−	2Myo-inositol gain (dimerization)
	C_24_H_40_NaO_18_	639.0	−	2Myo-inositol + Cyclohexane + 2OH + Na gain (dimerization/hydroxylation/adduct with metal)
	C_24_H_39_Na_2_O_18_	661.0	−	2Myo-inositol + Cyclohexane + 2OH + 2Na gain (dimerization/hydroxylation/adduct with metal)
**D-glucose**	* C_7_H_13_O_6_	193.0	+	=CH_2_ gain
	* C_6_H_12_O_8_	212.9	+	2OH gain (hydroxylation)
	C_12_H_16_O_8_	288.3	+	Benzene + 2OH gain (hydroxylation)
	C_12_H_22_O_9_	310.0	+	Cyclohexane + 3OH gain (hydroxylation)
	C_12_H_22_NaO_8_	317.0	+	Cyclohexane + H_2_O + Na + O gain (hydration/adduct with metal/oxidation)
	C_12_H_20_NaO_9_	331.9	+	Cyclohexane + H_2_O + Na + 2O gain (hydration/adduct with metal/oxidation)
	C_13_H_23_O_10_	339.0	+	Cyclohexane + 3OH + H_2_O + CH_2_ gain (hydroxylation/hydration)
	* C_7_H_13_O_8_	225.0	−	CH + 2OH gain (hydroxylation)
	C_10_H_19_O_11_	315.4	−	D-glucose gain − 2CH_2_ (dimerization, cross-ring cleavage)
	C_12_H_17_O_10_	321.4	−	D-glucose gain − OH − 3H loss (dimerization/dehydroxylation/dehydrogenation)
	C_18_H_31_O_13_	455.1	−	D-glucose + Cyclohexane + 2OH gain (dimerization/hydroxylation)
	C_24_H_31_O_21_	655.9	−	3D-glucose gain − 10H loss (dimerization/dehydrogenation)
	C_24_H_35_O_21_	659.9	−	3D-glucose gain − 6H loss (dimerization/dehydrogenation)
	C_24_H_41_O_21_	665.9	−	3D-glucose gain (dimerization)
**D-fructose**	* C_6_H_8_O_3_	128.1	+	3OH − 4H loss (dehydroxylation /dehydrogenation)
	C_6_H_4_O_6_	172.0	+	8H loss (dehydrogenation)
	* C_7_H_13_O_6_	193.0	+	CH_2_ gain
	C_10_H_18_O_7_	250.9	+	Tetrahydrofuran gain
	C_11_H_20_O_7_	264.0	+	Tetrahydrofuran + CH_3_ gain
	C_11_H_20_O_8_	280.0	+	Tetrahydrofuran + CH_2_OH gain
	C_11_H_20_O_9_	296.0	+	Tetrahydrofuran + CH_2_OH + OH gain (hydroxylation)
	C_12_H_18_O_11_	338.0	+	D-fructose gain − 4H loss (dimerization/dehydrogenation)
	C_12_H_18_O_13_	370.9	+	D-fructose + 2OH gain − 4H loss (dimerization/hydroxylation/dehydrogenation)
	C_17_H_29_O_13_	441.2	−	3D-fructose gain − 3OH − C loss (dimerization/dehydroxylation)
	C_17_H_29_O_14_	457.1	−	3D-fructose gain − 2OH − C loss (dimerization/dehydroxylation)
	C_18_H_27_O_15_	483.1	−	3D-fructose gain − OH − 4H loss (dimerization/dehydroxylation/dehydrogenation)
	C_23_H_38_NaO_18_	625.0	−	4D-fructose + Na gain − 3OH − CH_2_ loss (dimerization/adduct with metal/ dehydroxylation)
	C_24_H_38_NaO_19_	653.0	−	4D-fructose + Na gain − 2OH loss (dimerization/adduct with metal/ dehydroxylation)
	C_24_H_37_O_21_	661.9	−	4D-fructose gain − 4H loss (dimerization/ dehydrogenation)
**D-galactose**	C_6_H_4_O_6_	172.0	+	8H loss (dehydrogenation)
	C_12_H_22_O_6_	262.0	+	Cyclohexane gain
	C_12_H_18_O_8_	284.3	+	Benzene + 2OH gain − 10H loss (hydroxylation/dehydrogenation)
	C_12_H_22_O_8_	294.0	+	Cyclohexane + 2OH gain (hydroxylation)
	C_12_H_22_O_9_	310.0	+	Cyclohexane + 3OH gain (hydroxylation)
	C_12_H_20_NaO_9_	331.0	+	Cyclohexane + H_2_O + Na + O gain (hydration/adduct with metal/reduction)
	C_13_H_23_O_10_	339.0	+	Cyclohexane + 3OH + H_2_O + CH_2_ gain (hydroxylation/hydration)
	* C_9_H_13_O_7_	233.8	−	C_3_HOH gain
	* C_10_H_19_O_7_	251.7	−	C_4_H_7_OH gain (cross-ring cleavage)
	C_12_H_17_O_10_	321.3	−	D-galactose gain − OH − 4H loss (dimerization/dehydroxylation/dehydrogenation)
	C_12_H_12_NaO_12_	371.3	−	D-galactose + Na + O gain − 6H loss (dimerization/adduct with metal/reduction/ dehydrogenation)
	C_18_H_31_O_13_	455.2	−	D-galactose + Cyclohexane + 2OH gain (dimerization/hydroxylation)
	C_18_H_32_O_14_	472.2	−	D-galactose + Cyclohexane + 3OH gain (dimerization/hydroxylation)
	C_24_H_31_O_21_	655.1	−	3D-galactose gain − 10H loss (dimerization/dehydrogenation)
	C_24_H_37_O_21_	661.1	−	3D-galactose gain − 4H loss (dimerization/dehydrogenation)
	C_24_H_41_O_21_	665.0	−	3D-galactose gain (dimerization)

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
