# Peer review of "New Methodology for the Identification of Metabolites of Saccharides and Cyclitols by Off-Line EC-MALDI-TOF-MS"

_ijms, 2020, doi:10.3390/ijms21155265_

Round 1
Reviewer 1 Report
General comments:
The authors in this work carry out a combination of electrochemistry and MALDI-TOF technology for the determination of biological active compounds, as saccharides and cyclitols, demonstratingfor the first time the potential of the combination of electrochemistry and mass spectrometry analysis for the generation and detection of electrochemical product from saccharides and cyclitols. I think it is a well structured work in a technically point of view, however the exposition of some concepts, which deserve more attention, should be improved. Below some suggestion are listed:
- Regarding Material and methods section, more information on EC technique should be improved e.g. used voltage.
- Line 97 “Supplementary 1” is cited: I don't know if it is a typing error or I have not received the additional materials for viewing.
- An explanation that elucidates to the reader the reason why these types of compounds have been chosen for the development of the method should be made.
- The statistical process developed by the authors to describe the results is certainly very interesting and elaborate. But, in my opinion, while a functional analysis (for example two clinical groups in comparison) makes the reading of a heatmap, of a network almost obvious, the understanding of figures 7-8-9 in this case (in which deals with methodological parameters) it was really complex. The authors must accompany each figure with a more "didactic" description that can help the reader understand the results.
- Following this way, I believe (if the journal allows it) that the division of Results and Discussion sections is necessary to improve the understanding of the results.
- Reading the work, it is clear that Discussions and therefore the Conclusions section is really lacking. This makes it difficult to identify the initial objectives of the work, and consequently those achieved by scientific evidence.
- A section where the authors show or explain the possible applications of this method would be really important.
- Abbreviations are mentioned for the first time in the section of materials and methods, which is the final part of the work. Please adjust abbreviations the first time they occur in the text.
- Finally, some sentences are very long and structurally complex. A linguistic revision of the text is required.
Author Response
Response to Reviewer 1 Comments
Ref. No.: ijms-861844
New methodology for identification of metabolites of saccharides and cyclitols
by off-line EC-MALDI-TOF-MS
We are very grateful to your critical comments and thoughtful suggestions. Based on these comments and suggestions, we have made a careful revision of the original manuscript. A revised manuscript has been submitted, in which the modified sections are highlighted in red. Thank the Editor and Reviewers again, who made great contributions to improve our paper.
Point 1: Regarding Material and methods section, more information on EC technique should be improved e.g. used voltage.
Response 1: Authors are thankful for this thoughtful suggestion, additional information on applied electrochemical technique has been added to 'Materials and Methods' section as it was recommended by Reviewer.
Point 2: Line 97 “Supplementary 1” is cited: I don't know if it is a typing error or I have not received the additional materials for viewing.
Response 2: The Reviewer comment definitely has sense. The phrase 'Supplementary 1' (line 97) was deleted from the manuscript text.
Point 3: An explanation that elucidates to the reader the reason why these types of compounds have been chosen for the development of the method should be made.
Response 3: Authors are grateful for this interesting comment. This information has been added to the text. Cyclitols and sugars as plant bioactive compounds were selected for this study since representatives of both of these species have similar structural features and also play opposite activities in Diabetes Mellitus. Mechanisms of action of cyclitols as potential anti-diabetic compound are still understood poorly. In addition, the literature on potential metabolites of cyclitols is scarce. Selected analytes are well-known representatives of the species of sugars and cyclitols and it was interesting to observe similarities and differences in candidate metabolites between the species.
Point 4: The statistical process developed by the authors to describe the results is certainly very interesting and elaborate. But, in my opinion, while a functional analysis (for example two clinical groups in comparison) makes the reading of a heatmap, of a network almost obvious, the understanding of figures 7-8-9 in this case (in which deals with methodological parameters) it was really complex. The authors must accompany each figure with a more "didactic" description that can help the reader understand the results.
Response 4: Regarding the relevant issue highlighted by the Reviewer, the authors added sentences dedicated to more clear explanation regarding the interpretation of the presented graphs.
Point 5: Following this way, I believe (if the journal allows it) that the division of Results and Discussion sections is necessary to improve the understanding of the results.
Response 5: Thank you for this critical comment. Since the paper aimed at comparison of potential metabolites between the species, it could probably be more convenient to the readers to keep results and discussion together. The results and drawn discussions are presented point by point trying to following logical order and the final part regarding exploratory data analysis provides an overview of the whole findings. In addition, authors added more discussion and clarified some of the points of the results as it was recommended by the Reviewer. Therefore, now we hope that the paper is in more adequate form to be understandable for readers.
Point 6: Reading the work, it is clear that Discussions and therefore the Conclusions section is really lacking. This makes it difficult to identify the initial objectives of the work, and consequently those achieved by scientific evidence.
Response 6: Thank you so much for your critical remark. In attention to this valuable comment, sentences clarifying the aims of the study were added to Abstract and Introduction sections. Besides that, Discussion and Conclusions sections were incremented, in order to provide a better connection being proposed goals and the achieved results.
Point 7: A section where the authors show or explain the possible applications of this method would be really important.
Response 7: Thank you very much for this comment, which is indeed critical. The propositions for future applications of the method were added to the text.
Point 8: Abbreviations are mentioned for the first time in the section of materials and methods, which is the final part of the work. Please adjust abbreviations the first time they occur in the text.
Response 8: Authors are grateful for this comment. The abbreviations were adjusted upon their first appearance in the text.
Point 9: Finally, some sentences are very long and structurally complex. A linguistic revision of the text is required.
Response 9: Authors appreciate the comment of the reviewer. The manuscript text has been checked by english native speaker, which was helpful to improve the quality of the paper. Final text was also checked and corrected to improve the understanding of the most complex sentences. Authors hope that at the moment english of the manuscript meets the high standards of the journal.
Reviewer 2 Report
The paper describes a methodology for the characterization of selected low molecular weight analytes by combination of electrochemistry and MALDI TOF MS. The manuscript is well written and can be accepted after addressing very minor comments:
- Some parts of the text are wordy and repeatedly explain the abbreviations used already earlier - e.g., the first sentence of the Results and Discussion section.
- The experiments were performed with model substances. It would be useful to provide a brief section of future practical applications of the presented method.
- Correct the sentence on page 2 line 68
Author Response
Response to Reviewer 2 Comments
Ref. No.: ijms-861844
New methodology for identification of metabolites of saccharides and cyclitols
by off-line EC-MALDI-TOF-MS
We are very grateful to your critical comments and thoughtful suggestions. Based on these comments and suggestions, we have made a careful revision of the original manuscript. A revised manuscript has been submitted, in which the modified sections are highlighted in red. Thank the Editor and Reviewers again, who made great contributions to improve our paper.
Point 1: The paper describes a methodology for the characterization of selected low molecular weight analytes by combination of electrochemistry and MALDI TOF MS. The manuscript is well written and can be accepted after addressing very minor comments: Some parts of the text are wordy and repeatedly explain the abbreviations used already earlier - e.g., the first sentence of the Results and Discussion section.
Response 1: Authors are very grateful for appreciation of our work and for dedication of time and efforts to review our manuscript.
Point 2: The experiments were performed with model substances. It would be useful to provide a brief section of future practical applications of the presented method.
Response 2: The comment on the paper is definitely critical. Propositions for practical applications of the method have been added to the text.
Point 3: Correct the sentence on page 2 line 68
Response 3: Authors appreciate the given comment. The sentence on page 2 (line 68) has been corrected.
Reviewer 3 Report
The manuscript by Sagandykova et al. reports the potential metabolites that can be found after submitting some cyclitols and suggars to electrochemical transformation as detected by MALDI-ToF-MS. The metabolites arising from the electrochemical process are suggested to be potential metabolites that could be found after biological cell metabolism. Even though the data provided in this study may be of interest for metabolomic studies, there are some issues that should be addressed:
- If the methology proposes an off-line EC treatment, the nomenclature EC-MALDI-ToF-MS is not a proper one.
- Lines 364-365 (and over all the text): nomenclature for ions is not the correct one and lead to confusion. For example, it can be read “The fragment m/z 274 from D-pinitol may 363 result from m/z 387 [2M - H]+, -2CH2O/-3H2O.” Are the authors referring to the aduct [2M-2CH2O-H]+, or to what?
- Lines 420-429: what is the meaning of collecting two fractions? It does not make sense to collect only two fractions if the aim is to have few metabolites to increase the sensitivity of the analysis.
- Similarly, it does not make sense to provide data in figures 7 & 8 for the HCCA matrix in the MALDI-ToF-MS analysisis after the authors claim this matrix did not provide good results as compared to the other matrix used in the study, that is DHB.
- Lines 252-269: metabolites detected after positive ionization for cyclitols are compared with metabolites detected after negative ionization for suggars. There may not be any correlation between metabolites from both ionization methods.
- There is the sensation over the manuscript’s text that adducts from MS are not properly distinguished from the potential metabolites with biological value arising from the EC. For example, it is difficult to assume that dimers or aducts formed after gaining a phenol group may be relevant metabolites, or even exist, as a consequence of the cell metabolism (see Table 1 and lines 402-404). The same can be said for adducts resulting from the loss of 8 protons or gaining other chemical groups that are likely to come from the matrix in the MALDI-ToF-MS analysis.
- Figures 1-6: as commented in the precedent point, and in spite of the color code used, to clearly indicate which transformations are proposed to be generated in the EC process and which of them result in the MS analysis is mandatory in order to know the metabolites that could be relevantly associated to a physiological process.
- Please specify which metabolites were used for the statistical analysis in figures 7-9. Were they used the metabolites considered to be physiologically relevant, the metabolites arising from the EC transformation, or all the metabolites as detected by MALDI-ToF-MS?
Author Response
Response to Reviewer 3 Comments
Ref. No.:ijms-861844
New methodology for identification of metabolites of saccharides and cyclitols
by off-line EC-MALDI-TOF-MS
We are very grateful to your critical comments and thoughtful suggestions. Based on these comments and suggestions, we have made a careful revision of the original manuscript. A revised manuscript has been submitted, in which the modified sections are highlighted in red. Thank the Editor and Reviewers again, who made great contributions to improve our paper.
Point 1: The manuscript by Sagandykova et al. reports the potential metabolites that can be found after submitting some cyclitols and suggars to electrochemical transformation as detected by MALDI-ToF-MS. The metabolites arising from the electrochemical process are suggested to be potential metabolites that could be found after biological cell metabolism. Even though the data provided in this study may be of interest for metabolomic studies, there are some issues that should be addressed: If the methology proposes an off-line EC treatment, the nomenclature EC-MALDI-ToF-MS is not a proper one.
Response 1: Thank you so much for your critical Remark given for presented paper. Indeed, authors proposed the approach based on the combination of EC treatment and MALDI-TOF-MS in off-line mode. The notation for the approach such as EC-MALDI-TOF-MS is indeed inappropriate, however authors presented the notation such as 'off-line EC-MALDI-TOF-MS' that demonstrates the absence of the on-line connection between EC and MALDI-TOF-MS. Moreover, such notations are quite common for e.g. in case of off-line LC-MALDI-MS as it is stated in the following papers in respected journals on analytical chemistry (Chen, H.S.; Rejtar, T.; Andreev, V.; Moskovets, E.; Karger, B.L. Anal. Chem. 2005, 77, 2323–2331; Juhasz P, Lynch M, Sethuraman M, Campbell J, Hines W, Paniagua M, et al. J Proteome Res. 2011, 10(1), 34–45; Larsen TR, Bache N, Gramsbergen JB, Roepstorff P. J Am Soc Mass Spectrom. 2011, 22(6), 989–96). In addition, authors have corrected the notation 'EC-MALDI-TOF-MS' in the text with the addition of 'off-line'.
Point 2: Lines 364-365 (and over all the text): nomenclature for ions is not the correct one and lead to confusion. For example, it can be read “The fragment m/z 274 from D-pinitol may 363 result from m/z 387 [2M - H]+, -2CH2O/-3H2O.” Are the authors referring to the aduct [2M-2CH2O-H]+, or to what?
Response 2: Authors appreciate the comment given by Reviewer. The notation in the text [2M - H]+, -2CH2O/-3H2O is referring to the loss of two molecules of CH2O and three molecules of water from the initial [2M - H]+. The mentioned notation as well similar ones in the text has been corrected to clarify it for readers. Thank you so much for attracting our attention to this issue. We believe that corrected version of the notations helped us to improve the quality of our manuscript.
Point 3: Lines 420-429: what is the meaning of collecting two fractions? It does not make sense to collect only two fractions if the aim is to have few metabolites to increase the sensitivity of the analysis.
Response 3: Authors are grateful for this interesting comment. Samples after electrochemical treatment were collected by two fractions in order to decrease the volume of each fraction and analyse metabolites even those at low concentrations. Moreover, it was interesting for us to observe the difference in time by comparison of the intensity of the peaks between the fractions. However, this part of discussion has not been included to the text since we observed almost no differences between the fractions.
Point 4: Similarly, it does not make sense to provide data in figures 7 & 8 for the HCCA matrix in the MALDI-ToF-MS analysisis after the authors claim this matrix did not provide good results as compared to the other matrix used in the study, that is DHB.
Response 4: Authors appreciate the comment given by the Reviewer. Indeed, we studied two matrices for analysis of potential metabolites by MALDI-TOF-MS to observe the effect of the matrix and ionization mode for selection optimal parameters for the approach. Although the use of DHB matrix presented to be more suitable for the aimed purposes, the authors consider that useful information can be drawn from a comparison with the profiles of MS spectra obtained using HCCA, contributing to a more rich discussion of the results. For example, most unique and shared fragments obtained by using HCCA are also elucidated and main characteristics of acquired spectra are underlined. Such information may also be useful for researchers intending to reproduce the presented analysis and have different type of matrix available.
Point 5: Lines 252-269: metabolites detected after positive ionization for cyclitols are compared with metabolites detected after negative ionization for suggars. There may not be any correlation between metabolites from both ionization methods.
Response 5: Authors would like to thank the Reviewer for this valuable comment. May the text presented itself not very clear, but the authors did not aim to compare the fragments obtained in positive mode with those obtained in negative mode. The paired comparison refers to signals obtained before and after electrochemical process, individually, for each group of studied analytes. Alterations in the text were applied in order to clarify this issue following the comment of the reviewer.
Point 6: There is the sensation over the manuscript’s text that adducts from MS are not properly distinguished from the potential metabolites with biological value arising from the EC. For example, it is difficult to assume that dimers or aducts formed after gaining a phenol group may be relevant metabolites, or even exist, as a consequence of the cell metabolism (see Table 1 and lines 402-404). The same can be said for adducts resulting from the loss of 8 protons or gaining other chemical groups that are likely to come from the matrix in the MALDI-ToF-MS analysis.
Response 6: Authors would like to thank the Reviewer for this interesting remark, which is indeed reasonable. The manuscript aimed to elucidate the formation of all most relevant fragments presented in the obtained spectra, also intending to distinguish matrix-related fragments from others (which would configure as potential metabolites). The authors do agree that fragments generated by association with units coming from the matrix cannot be considered as candidate metabolites of the studied analytes since they are less likely to be formed in a human body. In order to clarify this issue, sentences stating what was aforementioned were introduced in the Results and discussions section.
Point 7: Figures 1-6: as commented in the precedent point, and in spite of the color code used, to clearly indicate which transformations are proposed to be generated in the EC process and which of them result in the MS analysis is mandatory in order to know the metabolites that could be relevantly associated to a physiological process.
Response 7: Authors are grateful for this comment. Transformations that might occur after the electrochemical treatment were proposed in Table 1. Indeed, it is important to distinguish the signals coming from proposed metabolites from those coming from MS analysis. Therefore, color codes for MS signals were used to make relevant comparison between control and electrochemically treated to build further discussion. Moreover, in attention to the issue arising from this relevant comment, a signaling was used in the Table 1, in order to highlight those fragments that can be interpreted was potential metabolites of the studied compounds. Besides that, “Potential metabolites” was switched to “Fragments” and “Proposed metabolic reaction” to “Proposed reactions”.
Point 8: Please specify which metabolites were used for the statistical analysis in figures 7-9. Were they used the metabolites considered to be physiologically relevant, the metabolites arising from the EC transformation, or all the metabolites as detected by MALDI-ToF-MS?
Response 8: Authors would like to express their gratitude for this critical comment on our manuscript. To avoid misunderstanding and to exclude the peaks that could arise as artifacts of MS analysis, the most intensive signals from spectra were selected for statistical analysis (Figures 7-9). Analysis of ionization of cyclitols and thus, artifacts of the analysis with mentioned matrices by MALDI-TOF-MS was carried out by our research group previously (Al-Suod H, Pomastowski P, Ligor M, Railean-Plugaru V, Buszewski B. Phytochem Anal. 2018; 29(5):528–37). Since MALDI analysis yet has 'blind spots' and not everything is clear till the very end on mechanisms of ionization of analytes, there is a chance that artifacts of the analysis (for e.g. as a result post-source and in-source decay) were included to statistical analysis. However, it is quite low since only the most intensive signals were included and such fragments are usually expected to have low abundance in MS spectra. In addition, MALDI is one of the most soft ionization techniques and the per centage of fragmentation during the analysis is not much high over the initial analytes. Finally, such artifacts of MALDI-TOF-MS analysis as adducts are not counteracting the identification of fragments after their conversion in electrochemical cell. We believe that proposed approach can find its application in analysis of candidate metabolites of sugars and cyclitols. Also a sentence explaining this point more clearly was introduced in Materials and Methods section (sub item “3.5. Exploratory data analysis”). Moreover, Table 1 has been revised to add more clarity on formed adducts as artifacts of the analysis and proposed metabolic reactions that might occur to analytes. All of the fragments that were reported in Table 1 could be proposed as physiologically relevant due to similarities in reactions in electrochemical cell and oxidative phase I cell metabolism, but the authors highlight that combination of EC cell with MS is only a simulation that allows to model the reactions that might occur in the human body. Moreover, some of the compounds marked with asterisk were assumed to be more likely formed in a human body since formation of adducts of fragments potentially show that they are less thermodynamically stable in a gas phase. Probably, it leads to suggestion that further rearrangements may occur in a human body, but it is still important to consider that the conditions in a human body (e.g. enzymatic reactions in case of metabolism process) differ significantly from thermodynamical stability in a gas phase. Therefore, the reported metabolites are only a proposition and can be referred as 'candidate metabolites'. The respective changes has been made in the manuscript text to clarify this issue.
Round 2
Reviewer 1 Report
The authors followed all the suggested indications, recalibrating results and discussion section.
Very interesting the additional information of table 1.
For me the work is ready for publication.
Reviewer 3 Report
The authors have conveniently addressed all the issues raised by the reviewers.
Table 1 should be checked for probable typing mistakes in lines 2 (C6H12O8) and 3 (C13H16O7): - 2OH gain? and - benzaldehyde gain?